# Cultural adaptation of the person-centered maternity care scale at governmental health facilities in Cambodia

Yuko Takahashi Naito[1]*, Rieko Fukuzawa[2], Patience A. Afulani[3], Rattana Kim[4], Hirotsugu Aiga[5]

1 Graduate School of Comprehensive Human Sciences, University of Tsukuba, Tsukuba, Japan, 2 Faculty of Medicine, University of Tsukuba, Tsukuba, Japan, 3 Departments of Epidemiology & Biostatistics & Obstetrics, Gynecology & Reproductive Sciences, University of California, San Francisco, California, United States of America, 4 National Maternal and Child Health Center, Phnom Penh, Cambodia, 5 School of Tropical Medicine and Global Health, University of Nagasaki, Nagasaki, Japan

* yukota619@gmail.com

**Data Availability Statement:** All relevant data are within the paper and its Supporting Information files.

## Abstract

### Background

In Cambodia, the importance of valuing women's childbirth experiences in improving quality of care has been understudied. This is largely because of absence of reliable Khmer tools for measuring women's intrapartum care experiences. Generally, cross-cultural development of those tools often involves translation from a source language into a target language. Yet, few earlier studies considered Cambodian cultural context. Thus, we developed the Cambodian version of the Person-Centered Maternity Care (PCMC) scale, by culturally adapting its original to Cambodian context for ensuring cultural equivalence and content validity.

### Methods

Three rounds of cognitive interviewing with 20 early postpartum women were conducted at two governmental health facilities in Cambodia. Cognitive interviewing was composed of structured questionnaire pretesting and qualitative probing. The issues identified in the process of transcribing and translating audio-recorded cognitive interviews were iteratively discussed among study team members, and further analyzed.

### Results

A total of 14 issues related to cultural adaptations were identified in the 31 translated questions for the Cambodian version of the PCMC scale. Our study identified three key findings: (i) discrepancies between the WHO recommendations on intrapartum care and Cambodian field realities; (ii) discrepancies in recognition on PCMC between national experts and local women; and (iii) challenges in correctly collecting and interpreting less-educated women's views on intrapartum care.

**Funding:** This work was supported by a grant-in-aid of the "YAMAJI FUMIKO NURSING RESEARCH FUND". The funders had no role in study design, data collection and analysis, decision to publish, or preparation of the manuscript.

**Competing interests:** The authors have declared that no competing interests exist.

## Conclusion

Not only women's verbal data but also their non-verbal data and cultural contexts should be comprehensively counted, when reflecting Cambodian women's intrapartum practice realities in the translated version. This is the first study that attempted to develop the tool for measuring Cambodian women's experiences during childbirth, by addressing cross-cultural issues.

## Introduction

Person-centered maternity care (PCMC) is highlighted in the WHO Quality of Care Framework for Maternal and Newborn Health, which composed of three dimensions: (i) effective communication; (ii) respect and dignity; and (iii) emotional support [1]. Recent evidence on women's experiences of care has been incorporated into the new WHO intrapartum care guidelines for a positive childbirth experience to enhance women-centered outcomes [2]. Earlier studies suggested that the person-centered dimensions determine women's decisions to seek health care [3–9]; their intentions to select the same facility for future deliveries [10,11]; their demands for more friendly care [3,12,13]; their satisfaction with quality of care [4,11,14]; and long-term impacts on their future reproduction [15,16].

Cambodia is a lower-middle-income country in South East Asia that has undergone significant hardship such as colonialization, civil wars, and genocide during the Khmer Rouge regime. Note that Cambodian health system was entirely damaged and even destroyed (incl. health infrastructures and skilled health workforce) during the hardship. Several earlier studies found that Cambodian women do not necessarily receive woman-centered care during childbirth at health facilities, and reported that women's poor perception towards the interpersonal aspect of quality of care is a significant barrier to health service utilization in Cambodia [6,17–19]. Women's inadequate utilizations of facility-based intrapartum care services led to delays not only in accessing life-saving medical interventions [9] but also in achieving SDG3 target [20]. Despite global recognition of the importance of quality of care, until recently, women's childbirth experiences had not been adequately reflected in the national guidelines for improving the quality of care [21]. Further, little is known about the extent of PCMC in Cambodia because no country-specific reliable tool to quantitatively assess women's experiences during childbirth is available.

There are various tools to measure women's childbirth experiences, but there is a lack of consensus on how to operationalize PCMC. To date, there is only one validated tool based on standard procedures of scale development that comprehensively captures the person-centered dimensions of the WHO Quality of Care Framework for Maternal and Newborn Health: the Person-Centered Maternity Care (PCMC) Scale [22].

The PCMC scale is a pre-validated tool to measure women's experiences of care during childbirth in health facilities, which was developed by Afulani et, al. and shown to have high content, construct, and criterion validity and good internal consistency reliability in Kenya and India, as described in detail elsewhere [22,23]. The original scale validated in Kenya includes 30 items on three key domains representing the "experience of care" dimensions of the WHO Quality of Maternal and Newborn Care Framework: (i) dignity and respect; (ii) communication and autonomy; and (iii) supportive care [1]. Response options are based on four-point frequency scale, i.e. "0 = No, never", "1 = Yes, a few times", "2 = Yes, most of the time", "3 = Yes, all the time". The full scale and subscales showed good internal consistency reliability in Kenya, India, and Ghana, with a Cronbach's α value of over 0.8 for the full scale and ranging between 0.61 and 0.75 for the subscales [24].

When developing a cross-cultural tool, multi-dimensional factors need to be considered not only in translation but in cultural interpretation, values, and attitudes [25]. Moreover, cross-cultural researches generally require strategies for adequately addressing the fact that a concept in a certain setting may not be identical or comparable to that in another setting and that a tool appropriate in one context could be inadequate in another context [26]. A questionnaire translated into another language must meet the following requirements: (i) be valid, reliable, and cost-effective; (ii) exhibit appropriate levels of semantic and conceptual equivalence [27]. In developing a cross-cultural tool involving translation from an original language to another language, simple forward and back translation techniques do not automatically maintain semantic equivalence across cultures [28]. Therefore, researchers responsible for cross-cultural tool development should assess not only semantic equivalence but also cultural equivalence between the original and translated tools, by using multiple quantitative and qualitative techniques [29]. However, few studies previously explored and described how culture and language influences the process of tool adaptation [30]. Note that majority of cross-cultural studies fail to perform adequate pretesting [31]. This leads to not only poor comparability but also poor validity and reliability [32].

The present study therefore aims to develop a reliable and valid Khmer version of the PCMC (Kh-PCMC) scale for use with Cambodian postpartum women. We adapted the pre-validated PCMC scale in Kenya, India and Ghana, and applied the standard procedures of cross-cultural translation and adaptation.

## Materials and methods

### Ensuring equivalence

Cross-cultural research requires equivalence, which refers to the degree to which survey measures or questions can assess identical phenomena across two or more cultures [33]. An original scale and its translated one need to be equally natural, acceptable, and feasible [31]. The focus is to achieve conceptual equivalence than procedural equivalence [34]. Cognitive interviewing is a useful approach for assessing conceptual equivalence of survey questionnaires [35]. To minimize a potential gap in conceptual equivalence, we utilized etic and emic anthropological model. Etic constructs exist in identical form across cultures, while emic constructs exist in a single culture [36]. The definition for each criterion and methods used in the study are presented in Table 1.

### Study site

This study was conducted at two governmental health facilities in Cambodia. One urban hospital in Phnom Penh and one rural health center in Kampong Chhnang province were selected

**Table 1. Equivalence criteria for cross-cultural translation and adaptation.**

| Criteria | Definitions | Process |
|---|---|---|
| Conceptual equivalence | Construct exist in two or more cultures and can be measured using similar or different survey questions | •Committee translation approach <br>•Expert reviews <br>•Cognitive interviewing |
| Semantic equivalence | Equivalence in the meaning of words, and achieving it may present problems with vocabulary and grammar | •Committee translation approach <br>•Expert reviews <br>•Cognitive interviewing |
| Content validity | The content of each item of the instrument is relevant to the phenomena of each culture being studied | •Literature review <br>•Content expert reviews <br>•Cognitive interviewing |

by convenience sampling, as this is a cross-cultural adaptation study [37]. The urban hospital provides both routine and emergency services including surgery and blood transfusion. Mean number of delivery cases at the hospital was reported as approximately 600 per month, of which 30% are caesarian deliveries. It serves women from a variety of clients (e.g. various domicile, diverse economic levels, different religions, and both low and high-risk pregnancies). The health center serves 10,000–20,000 population in its catchment area, and provides antenatal care, normal delivery, immunization, health education, and referral services [38]. Mean number of normal delivery cases at the health center was reported as approximately 10 per month.

## Translation into Khmer

The overall procedures followed the WHO guideline on translation and adaptation of instruments, which include six steps: (i) forward translation; (ii) expert review; (iii) back-translation; (iv) pretesting and cognitive interviewing; (v) final version; and (iv) documentation (Fig 1) [34]. Firstly, a formal permission of translation of PCMC scale into Khmer language was obtained from Dr. Afulani, the PCMC scale developer. Committee translation approach (Fig 2) was employed to ensure transparency and quality of translation [27] due to the limited professional translation resources for Khmer, one of minor languages. The PCMC scale was translated into Khmer, parallelly by two independent Khmer-English bilingual translators whose mother tongue is Khmer. The two versions were then synthesized by a Cambodian Khmer-English-Japanese trilingual linguistics expert, and reviewed by a Japanese Khmer-English-Japanese trilingual linguistics expert to generate the first version of Kh-PCMC scale. It was then further evaluated independently by eight Cambodian experts, of which two were Khmer monolinguals. Ambiguous and unmatched texts were thoroughly discussed and addressed among the study team members including the trilingual principal investigator whose mother tongue is Japanese, to generate the Kh-PCMC scale version 2.

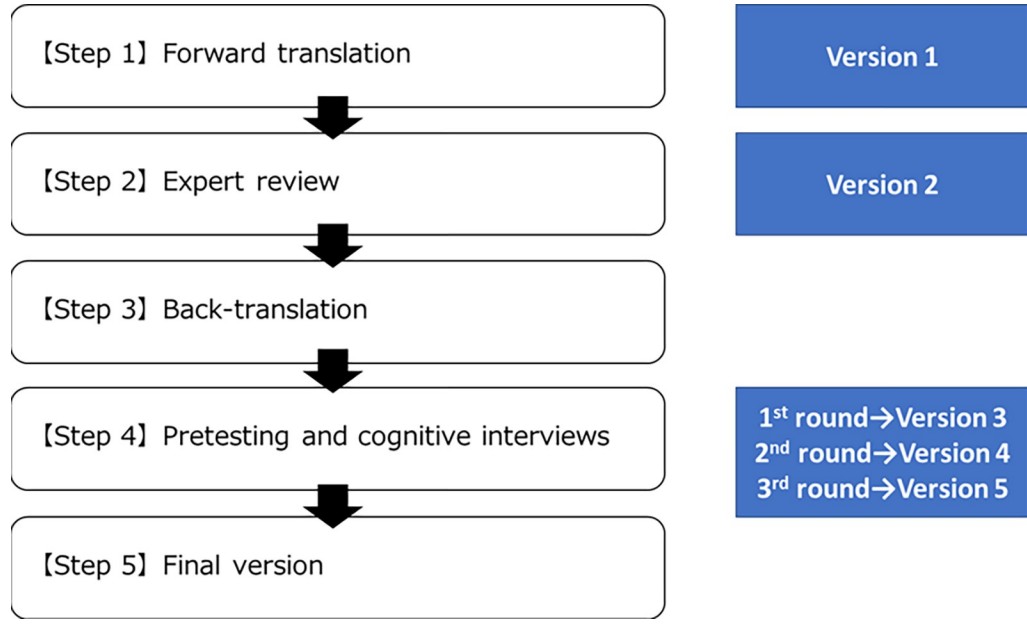

**Fig 1. Process of translation and adaptation of instrument.**

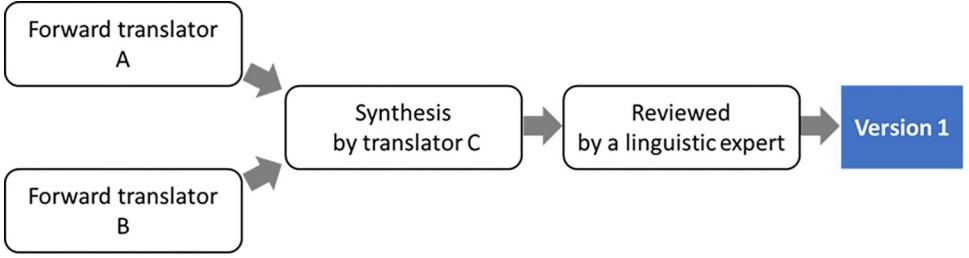

**Fig 2. Process of committee translation approach.**

## Cognitive interviewing

Cognitive interviewing is described as administration of a draft survey questions while collecting additional verbal information about survey responses is aimed at evaluating the quality of responses or determining whether questions generate the information a researcher intends [39]. Verbal probing approach [40] was conducted using cognitive interview guide developed by this study. Then, pre-developed proactive verbal probing was flexibly used according to the context settings and interviewees' personality: (i) What happened when?; (ii) How did you arrive at that answer?; (iii) Why do you say. . .?; (iv) What does [a key term] mean to you?; (v) Was this question difficult for you to answer?; and/or (vi) How would you rephrase this question to make it better? [22,41]. Reactive verbal probing was also used according to the respondent's reactions.

## Sampling and recruitment

Maximum variation sampling which is designed to produce diversity in types of individuals to be recruited [25] was employed for sampling postpartum women. This is because poor, less educated, younger, and minority are less likely to receive quality care [42]. While an earlier study recommended not more than 10 for each round [25] as the total number of respondents for cognitive interviewing, others recommended 10–30 [40] or at least 15 [43]. Therefore, this study recruited 20 postpartum women for cognitive interviewing. Eligibility criteria for sample were women aged 18–49 who gave births during the last seven days at the health facilities. Those having either undergone stillbirths or had their infants hospitalized due to serious delivery complication (e.g. congenital diseases and cerebral palsy) were excluded from the samples for this study. Participants in cognitive interviewing were purposively sampled from the delivery list available at health facilities. Both written informed consents to participate in cognitive interviewing and verbal agreements to audio-recording were obtained after verbal explanation of the study. Either baby soap or baby powder worth 10,000 Cambodian Riels (2.50 United States Dollar) was given to each participant, as appreciation for their participations.

## Data collection

Three rounds of cognitive interviewing with 20 postpartum women were undertaken until no new cognitive response errors produced among respondents during the period from January to March 2021 (Table 2). Three native Khmer-speaking female Cambodian nursing students were recruited and trained on cognitive interviewing skills necessary for interviewers. Each face-to-face online interview was conducted in Khmer using video-call software by the team composed of two interviewers and field assistants. One of the interviewers was a native Khmer speaking female Cambodian nursing student responsible for asking structured questions in Khmer. The other was the principal investigator who asked additional probes and took field notes. The Cambodian interviewer read aloud each question in the structured questionnaire to

**Table 2. Respondent sample.**

|  | 1st round | 2nd round | 3rd round | Total |
|---|---|---|---|---|
| Urban hospital | 7 | 4 | 4 | 15 |
| Rural health center | 3 | 1 | 1 | 5 |
| Total | 10 | 5 | 5 | 20 |

let women answer based on the best fits from the response options, and entered participant's response into the online questionnaire created by Google Form (Google LLC, California, United States). The principal investigator then asked additional probes for items that respondents appeared confused about or which they took a long time to respond, by applying the verbal probing approach [40].

## Data analysis

Audio-recorded interview data were first verbatim transcribed in Khmer and then translated into English and Japanese. For quality assurance, the principal investigator and Japanese trilingual linguistic expert independently reviewed the transcripts while listening to audio-recorded interviews. The transcripts and field notes were then reviewed by the principal investigator to identify ambiguous or confusing questions, and classified into the typologies of question failure using Appraisal System for Cross-National Survey [44] and Four-Stage Task Analytic Model [45]. The version was finalized by thoroughly addressing issues related to equivalence and building consensus among the study team members.

## Ethics

The study received ethical approval form the Ethics Committee, Faculty of Medicine, University of Tsukuba on 24 December, 2020: Reference: IRB1605, and National Ethics Committee for Health Research, Ministry of Health Cambodia on 30 December, 2020: Reference: #322 NECHR. Written informed consent was obtained from all participants, using informed consent form which was approved by the Ethics Committee, Faculty of Medicine, University of Tsukuba and National Ethics Committee for Health Research, Ministry of Health Cambodia.

## Inclusivity in global research

Additional information regarding the ethical, cultural, and scientific considerations specific to inclusivity in global research is included in the S1 Checklist.

## Results

### Characteristics of the sample

Twenty early postpartum women who were 2.9 days on average after childbirth were interviewed. The 20 women represented a variety of age group [mean: 28.5, range: 18–42 years of age], parity [mean: 1.9, range: 1–4], religion, occupation, socioeconomic backgrounds, maternal characteristics, and mode of delivery (Table 3).

### Issues identified

A total of 14 issues were identified in the 31 translated questions through the cognitive interviewing (Table 4). Of 14 issues, 12 (85%) were at comprehension stage in cognitive process [45]. These issues were categorized into five types by using the Appraisal system for Cross-National Survey [44]: (i) translation and adaptation; (ii) translation vocabulary; (iii) reference

**Table 3. Characteristics of respondents (n = 20).**

| Characteristics | | N | (%) |
|---|---|---:|---:|
| **Age (years)** | | | |
| | Mean [range] | 28.5 [18–42] | |
| | <20 | 2 | 10.0 |
| | 20–24 | 3 | 15.0 |
| | 25–29 | 7 | 35.0 |
| | 30–34 | 4 | 20.0 |
| | >35 | 4 | 20.0 |
| | Total | 20 | 100.0 |
| **Parity** | | | |
| | Mean [range] | 1.9 [1–4] | |
| | 1 | 9 | 45.0 |
| | 2 | 5 | 25.0 |
| | 3 | 4 | 20.0 |
| | 4 | 2 | 10.0 |
| | Total | 20 | 100.0 |
| **Marital status** | | | |
| | Married | 20 | 100.0 |
| **Religion** | | | |
| | Khmer | 16 | 80.0 |
| | Khmer Muslim | 3 | 15.0 |
| | Cristian | 1 | 5.0 |
| | Total | 20 | 100.0 |
| **Occupation** | | | |
| | Farmer | 4 | 20.0 |
| | Factory worker | 7 | 35.0 |
| | Housewife | 6 | 30.0 |
| | Self-employed retail | 2 | 10.0 |
| | Company employee | 1 | 5.0 |
| | Total | 20 | 100.0 |
| **Education** | | | |
| | No school education | 2 | 10.0 |
| | Primary | 8 | 40.0 |
| | Lower secondary | 6 | 30.0 |
| | Higher secondary | 2 | 10.0 |
| | University | 2 | 10.0 |
| | Total | 20 | 100.0 |
| **Economic status** | | | |
| | Non poor | 16 | 80.0 |
| | Poor | 4 | 20.0 |
| | Total | 20 | 100.0 |
| **Mode of delivery*** | | | |
| | Normal | 12 | 60.0 |
| | C/S | 8 | 40.0 |
| | Total | 20 | 100.0 |

* One vacuum delivery was excluded due to neonatal outcome. Forceps is not practiced in Cambodia.

**Table 4. Issues identified from cognitive interviewing.**

| Cognitive process* | Feature | Potential problem** | # | Item title | Original Question | Revised questions |
|---|---|---|---|---|---|---|
| Comprehension | Translation/ adaptation | Words requiring adaptation | 2 | introduce themselves | During your time in the health facility did the doctors, nurses, or other health care providers **introduce themselves** to you when they first came to see you? | During your time in the health facility did the medical staffs **introduce themselves** to you when they first came to see you? **For example, their name or profession.** |
| | | | 3 | being called by name | Did the doctors, nurses, or other health care providers **call you by your name**? | Did the medical staffs **call you by your name**? Did the medical staffs **call you appropriately**? |
| | | | 8 | involvement in decision | Did you feel like the doctors, nurses or other staff at the facility **involved you in decisions** about your care? | Did you feel like the medical staffs at the facility **considered your ideas in decisions** about your care? For example, can you decide for yourself whether you want to have a natural or caesarean section? |
| | | | 10 | delivery position choice | During the delivery, do you feel like you were able to **be in the position of your choice**? | During the delivery, do you feel like you were able to be in your **favorite free position**? |
| | | cultural confusion | 19 | when you needed help from medical staff | **When you needed help**, did you feel the doctors, nurses or other staff at the facility **paid attention**? | When you needed help, did you feel the medical staffs at the facility **respond to what you need**? |
| Comprehension | Vocabulary | Technical term | 10 | delivery position choice | During the delivery, do you feel like you were able to be in the **position** of your choice? | During the delivery, do you feel like you were able to be **in your favorite free position**? |
| | | Multiple definitions | 14 | talk about feeling | Did the doctors and nurses at the facility talk to you about how you were **feeling**? | Did the medical staffs at the facility talk to you about how you were **feeling (Physical/ Psychological)**? |
| Comprehension | Reference points | missing | 1 | time to care | How did you feel about the amount of **time you waited**? Would you say it was very short, somewhat short, somewhat long, or very long? | Did you feel to **wait long or short from when you arrived to when you received care**? |
| | | Lack of understanding of respondents | 6 | visual privacy | During examinations in the labor room, were you covered up with a cloth or blanket or screened with a curtain so that you did not feel exposed? | During examinations in the labor room (**for example, pelvic examination**), were you covered up with a cloth or blanket or screened with a curtain? |
| | | Vague | 9 | consent to procedures | Did the doctors, nurses or other staff at the facility ask your permission/ consent before **doing procedures on you**? | Did the medical staffs at the facility ask your permission/consent before doing procedures on you? **For example, pelvic examination and episiotomy** |
| | | | 12 | explain procedures | Did the doctors and nurses explain to you why they were **doing examinations or procedures on you**? | Did the medical staffs explain to you the objectives or reasons why they were doing examinations or procedures on you? **For example, pelvic examination or fetal heart rate monitoring** |
| Comprehension | Translation/ adaptation | Uncommon expression | 7 | record confidentiality | Do you feel like your health **information** was or will be kept confidential at this facility? | Do you feel like your health information was kept confidential at this facility? **For example, the information on the medical record.** |
| Retrieval | Task performance | Non reachable answers | | | | |
| Response | Response category | Nonexhaustive | | | | |

* Four stages task analytic model (Tourangeau, 1988 [45]).

**The Appraisal system for Cross-National Survey (Lee, 2014 [44]).

points; (iv) non reachable answers; and (v) response category. Furthermore, it was found that six of those 12 issues at comprehension stage required cultural adaptation. S1 Table presents the process of translation into Khmer and the reasons for revisions.

**Cultural translation and adaptation.** The most common issue identified through cognitive interviewing was the issue of cultural translation and adaptation which needs to be considered in the cultural context.

For instance, Item #2 "introduce themselves": the initial Khmer translation reflected the expert's suggestions to use "welcome" instead of "introduce themselves," as Cambodian people normally did not introduce themselves. But this did not seem semantically equivalent. At the urban hospital, three of 10 respondents at the first round described how they were accepted or guided at the reception upon their admission. On the other hand, at the rural health center, midwives and women were mutually acquainted as members of the same community. For this reason, midwives did not introduce themselves to women, when they visited the health center. The results of qualitative probes however revealed that nine of 20 respondents wanted to know which midwife would be responsible for their childbirths so they could call them when there is a problem. Thus, we re-translated "welcome" back to "introduce themselves" and added an explanation. The response errors were addressed by the additional explanations.

Similarly, Item #3 "being called by name" was replaced by "called appropriately" in the translated based on the experts' suggestions. But it was found that 11 of 20 respondents were called by name when getting injections and examinations at the urban hospital. Surprisingly, two respondents at the urban hospital stated that they would not have minded even if they had been called by their room numbers or bed numbers, because they could be identified by that (ID2, ID10). Three of 20 respondents stated that being called by name was more appropriate because "*There are many patients, there is no mistake* (ID6)" and because "*I have my name, so it is natural to be called by name* (ID13)". On the other hand, being called by name could be interpreted as an impolite or inappropriate way, under the condition where midwives and women were long-time acquaintances in the rural community. Four of 20 respondents stated that being called by either *Bong* (prefix for older woman in Khmer) or *Oung* (prefix for younger woman in Khmer) was a more comfortable way (ID9, ID15, ID17, ID21). The rest of the women stated that being called both ways was okay. Because "called appropriately" is attributed largely to individual preference, we decided to use both the original question ("called by name") and the context-specific question ("called appropriately") as the items for the survey, to assess the level of appropriateness of the two.

Regarding Item #8 "involvement in decision", in the process of identifying several translation options from qualitative probes, we found that the response errors in "involvement in decision" were not attributable to vocabulary-related issues but rather lack of familiarity with the concept. Eleven of 20 women who had vaginal deliveries were unfamiliar with the intention of the question, probably because they had never thought or participated in decision-making about their care. Two respondents stated "*I am happy to follow the doctors' recommendations*" (ID3, ID8). Another respondent stated "*Following midwives' recommendations was important, because it would lead to good results.*" (ID4). On the other hand, six respondents who had cesarean deliveries at the urban hospital stated "*I agreed to doctor's explanations and recommendations. Then, I decided to have cesarean delivery by myself*" (ID10, ID17). One highly educated woman (i.e. a university graduate) stated that the "*I wanted to give birth normally but I got surgery. I wanted to have more explanations for a need for cesarean delivery. . .. But, I couldn't. I decided to have surgery for my own life and my baby's life because no one takes responsibility if any problems happened*" (ID7). Overall, "involved in decisions" was interpreted as following doctors' and midwives' recommendations and blindly accepting them. Thus, we decided to add a scenario on whether or not an opportunity to choose the ways of care was given to a woman (Table 4). The response errors were addressed by the additional explanations.

Item #10 "delivery position choice" did not make sense among respondents because it is common to give birth in the supine position in Cambodia. Thus, "choice" was replaced by "favorite free position" that is more understandable to Cambodian women. It was found in qualitative probes that five of 20 women preferred free movements during labor to reduce labor pain and contribute to smoother delivery. One woman at the urban hospital reported that free delivery position was explained to her during delivery, by using its health education material (posters). As a result, she could freely move during labor. For this reason, "favorite free position" was employed as a replacement for "choice".

When responding to Item #19 "when you needed help from medical staff", three of 20 respondents at the urban hospital misinterpreted the item as need for help from their family members. We reminded them that the item implies help not from family members but from medical staff, when this misinterpretation occurred. By doing so, mothers' cognitive response errors were adequately addressed.

**Task performance and response category.** Regarding Item #7 "record confidentiality", 13 of 20 respondents stated that they did not know whether their health information was kept confidential. While, five of 20 respondents mentioned that no information was kept confidential (incl. household poverty specified in poverty ID card and HIV status). All the 20 respondents stated that they wanted to share their personal health information with their family members. In view of these statements, another response category "don't know if it was kept secret" was added as a response category.

**Women's cognitive issues.** The respondents appeared to have difficulties in selecting one out of multiple answer options in frequency-scale-based items. Although the interviewers read aloud each response option, the respondents often stated just "Yes" without selecting any specific option given. Women's response "Yes" were often inertial affirmative answers, implying either limited understanding or misinterpretation of the question intent. Women's initial response often needed to be reconfirmed by asking further supplementary qualitative probes. For example, one woman responded "Yes" as the initial response when being asked the question about record confidentiality. Yet, it found later that she did not even know whether the information was kept confidential. Thus, supplementary explanations were often necessary for some questions.

Moreover, it was found that many respondents found it difficult to correctly understand the questions on abstract concepts, for instance when asked the question "what respect means?". Many women mentioned that it was good enough that medical staff were ready to see them on time.

**Feasibility and applicability of PCMC scale in Cambodia.** Mean number of minutes spent on a cognitive interviewing was 65 minutes (range: 40–90 minutes). Overall, the length of interview time was acceptable to respondents, although some families complained that it was too long. All the 20 respondents preferred to have a face-to-face interview, though it was more time-consuming than self-administration of the questionnaire. Some stated "...*Because it is difficult for me to read through the questionnaire*..." and "...*Because I have never experienced in self-administering any questionnaire before*...". Those with no formal education took longer time than those with formal education. An example is that less educated women were neither experienced in nor familiar with selecting one of the four response options for the questions. One woman without formal education repeatedly stated "*Thank you*" without responding to the questions, because she was so delighted and honored to be interviewed for the first time. Since they usually communicate only in very basic Khmer, supplementary explanations were often necessary, e.g. by: (i) paraphrasing a question in easier words; (ii) giving specific examples; and (iii) repeating the question. Thus, face-to-face interview was the feasible mode of collecting data on care, when targeting Cambodian postpartum women.

## Discussion

This study described the process of cultural translation and adaptation of the PCMC scale to the Cambodian context. The team translation approach contributed to conducting a better-balanced translation despite the limited linguistic human resources [26]. Unsurprisingly, subsequent cognitive interviewing with respondents revealed significant response errors as evidence for the limitations of using only forward and back translations. A majority of the issues found in our cognitive interviewing were categorized into cultural adaptation [44].

Our study identified three key findings: (i) discrepancies between the WHO recommendations on intrapartum care [2] and Cambodian field realities; (ii) discrepancies in recognition on PCMC between national experts and local women; and (iii) challenges in correctly collecting and interpreting less-educated women's voices on intrapartum care. These were methodological difficulties in conducting a cross-cultural study targeting Cambodian women. To address these difficulties, we discuss the applicability of anthropological approach to reframing and unpacking existing methodologies for cross-cultural adaptation, by focusing on three perspectives.

### Discrepancies between the WHO recommendations and local realities

To address the discrepancies between the WHO recommendations and local realities, Cambodian cultural contexts need to be deeply considered. An attempt to seek local meanings, logics and interpretations is key to ensure questions based on the real understanding of concepts [46]. This study identified five types of discrepancies.

First, taking the concept of "self" as an example, Item #3 "being called by name" is based on Western sense of value derived from individualism. It can only capture the limited part of Asian concept of "self" that is derived more from collectivistic culture [27]. Cambodian populations have a unique culture concerning a personal name. Some respondents in our study stated that being misidentified with others needs to be avoided, even by being called by their room and/or bed numbers. This might be attributed simply to the greater number of patients at the urban hospital, not necessarily derived from the Cambodian way of respecting an individual. This result is in line with an anthropological study that Cambodian people believe the human being inseparably consists of physical body and name [47]. In Cambodian context, the body is a human being's container, while its name is given to the body. This inseparable unit identifies a "self" that distinguishes from others. One of the important foundations for correctly interpreting Cambodian contexts is Buddhism sense of self such as "reincarnation" and "karma". It is believed in Cambodia that every woman has eight reincarnation stages for eight rites of passage during lifetime from birth to death [47]. Childbirth is one of eight rites of passage for a woman. It is also believed that a woman's body at the current reincarnation stage will be reborn through childbirth and her name should be religiously changed when entering the next stage. As a result, Cambodian women might not be sensitive when being called by their legal names. Further, it is good harmony with Cambodian customs related to personal names: (i) Cambodian people sometimes change their names due to the results of fortune-telling; (ii) some differentiate their names between business use and private use; (iii) some children are given "school names" by teachers. Another possible explanation for not commonly being called by name is that Cambodian people are likely to call a senior "*Bong*" as a sign of respect and junior "*Oung*" as a sign of friendliness without using their family or given names. This way of calling others is important in Cambodia. This is consistent with the study in Kenya that Kenyan mothers would like to be called in a respectful manner [48]. In Cambodia, culture of respecting seniors is commoner in rural areas where community-based interpersonal relationship is deeper. In those rural areas, being called by name could make people feel emotionally distant and less respected.

Second, Item #8 "involvement in decision" can be made only where there are choices. The results of cognitive interviewing indicate that the question on decision-making on a cesarean delivery was likely to be interpreted, by Cambodian women, as an informed consent or unquestioned adherence rather than autonomous or joint decision with medical staff. Cambodian women rarely claim an alternative option proactively, hence have no expectations of being engaged in clinical dialogues. This may be potentially related to the concept of "karma" in the Buddhism fatalism, implying that people accept the situation no matter how irrational the way of caring is to minimize frictions and collisions with others [49]. The results of this study could be explained also by gender norm. This is consistent with an earlier study in India that reported male-dominant gender norm significantly influences decision-making process [41]. In fact, we encountered the situation where husbands refused to let their wives participate in this study in the consent process. This is largely because women's inability to participate in decision making is socially institutionalized. This is one of the significant discrepancies between globally emphasized importance of women's self-determination and local gender norm.

Third, Item #10 "delivery position choice" is also related to decision-making. Cambodian women who participated in cognitive interviewing simply accept supine position. This is because they have no doubt about giving birth in the supine position, as freestyle delivery has not been adequately known to women and practiced by medical staffs in Cambodia. This finding is consistent with several earlier studies that reported that: (i) women in Cambodia had no choice but to deliver in supine position [50]; (ii) 70% and 59% of women in rural Kenya and in Ghana respectively had no choice but delivery in supine position [24]; and (iii) only less than 5% of women in Malawi knew the positions other than supine position [51]. The latest WHO guidelines recommend upright rather than recumbent positions at the second stage of labor might reduce episiotomy and instrumental vaginal births [2]. A Cochran review reported that the benefits of non-supine positions include reduced duration of the second stage of labor and labor pain [52]. Supine position however remains dominant in many low- and middle-income countries. Thus, evidence-based local implementation needs to be accelerated in each existing health system.

Fourth, regarding Item #19 "when you needed help, did you feel the medical staffs at the facility respond to what you need?", some women were confused with the question about need for help by misinterpreting help from family members. This can be explained, to a large extent, by task sharing between doctors, midwives, nurses, and women's family members that were derived from shortage of health workforce. Note that Cambodia suffers shortage of health professionals, i.e. 1.15 physicians, nurses, and midwives per 1,000 population [53] against its global benchmark 4.45 [54]. Many non-invasive nursing care services (incl. bedside hygiene, bathing, and changing sanitary napkin) were provided primarily by patients' family members in Cambodia [55]. Confusion on the question "when you needed help" may be due to this context.

Fifth, it was found that Item #7 "record confidentiality" was often not a concern. For instance, even data on household's economic level and individual's HIV status were shared and disclosed. This finding was contrary to the results of other earlier studies in African countries with high HIV prevalence, where patients expect personally sensitive information such as HIV status to be strictly kept confidential [56,57]. This may imply what confidentiality means differ between women in those African countries and those in Cambodia. To clarify and deepen the understanding on confidentiality perceived by Cambodian women, a further study is required to understand.

Overall, the results of this study are consistent with findings elsewhere that some global concepts of PCMC may not resonate with local women's views and perceptions on PCMC

[41]. This is particularly so, where women are in a society noted for blind adherence to expert suggestion, implicit consent, poor awareness of alternative options, and gender norms and social hierarchy between medical staff and care receivers. Reframing, education, and adjustment in the local contexts are needed to implement WHO recommendations. Generally, use of abstract and conceptual terms in translation is recognized as one of the effective strategies for achieving semantic equivalence. Yet, it did not work well in Cambodia. Our study found that anthropological approach through qualitative probing more successfully captured the realities of Cambodian women's perceptions of PCMC [36].

## Discrepancies between Cambodian experts' and women's perceptions

It was found what and how Cambodian women have felt and experience during childbirth were not necessarily the same as Cambodian experts in reality. Changes made during translation suggested by the experts appeared sometimes semantically different from their original concepts. This finding on translation-related issue is supported by results of several earlier studies [30,58]. Robyn reported that discrepancies in understanding of concepts between experts and women are often attributed to their differences in cultural perspectives, knowledge, and sense of logical thinking [46]. The experts for the study evaluated some items such as Item #3 "being called by name" as "not relevant in Cambodian context", because they were not commonly practiced in Cambodia. However, Afulani, who developed the PCMC scale, mentioned that "...*the fact that something is not normally done does not mean women don't want it. The PCMC questions should be aspirational to what women want and not just what is normally done...*". The experts should sincerely listen to lay-persons' voices despite their challenges in doing so, because lay-persons provide significant information that otherwise could be missed or overlooked [46]. To deepen understanding of mothers' views, the present study employed not only frequency-scale-based structured questions but also qualitative probing. The qualitative probes successfully provided insightful understandings on what matters to Cambodian women during childbirth, because this study placed greater weight on cognitive interviewing with the respondents than the expert review. Cognitive interviewing was optimized to minimize the response errors, improve equivalence and validity, and contribute to accurate and less biased results. Several recent studies highlighted greater contribution of cognitive interviewing to tool development [59–61] and concluded cognitive interviewing as an essential element in the process of developing a cross-cultural tool [62].

## Interpretation of less-educated women's views

Interpretation of less-educated women's views required a certain ingenuity. The issues identified through cognitive interviewing were more related to women's cognitive process than to lack of clarity in question texts. Because less-educated Cambodian women were not accustomed to participating in a survey, their first response to questions were not always reliable. For example, some respondents always answered "Yes" as if automatically, without seriously thinking about their response. This might be attributable to typical acquiescence bias among less-educated Cambodian women. Their responses to the subsequent qualitative probes often explained their way of thinking, e.g. why and how they reach the answers. This finding is consistent with an earlier study in India where the respondents were more comfortable when being requested to state their experiences narratively than when framing answers to the survey questions. The study reported that subsequent qualitative descriptions were sometimes contradictory to the responses to structured questions [41]. The frequency response options in the PCMC scale provides an opportunity to further explore respondents' initial response.

In addition, it was challenging to elicit rich qualitative data from less-educate women. We failed to elicit deeper meanings of "respect" despite repeated qualitative probes. This is probably because many respondents had difficulties in understanding abstract concept such as "respect". This finding supports the results of an earlier study in Cambodia that reported the inadequate evaluation of quality of care through measuring satisfaction with medical staff's attitude and health outcome among less educated women [63]. A similar challenge was reported from a study in Côte d'Ivoire where less-educated women's responses were not reliable enough because they were not accustomed to be asked questions on their health outcomes [64]. This can be explained by local women's poor expectations about health care quality, because they take low health care quality for granted and are used to it [65]. Thus, recognizing what women know about their realities narratively is the key first step to ensuring the solutions are based on real understanding of the issues and needs [46]. Putting women at the center of care often requires reframing. Reframing into local contexts is key to successful development of the Kh-PCMC scales.

To fully describe Cambodian women's realities, it is important to triangulate non-verbal data and description of the cultural context rather than relying on only verbal data from women. Multi-methods should be employed as the study design and methods. Anthropological approach such as "Participatory Rural Appraisal (PRA)" and "Participatory Learning and Action (PLA)" [66] would be useful, including direct observation of care, fieldwork, drawing and mapping, and interviews with other stakeholders such as family and companions. The present study, therefore, highlight the significance of understandings on invisible local cultural contexts using multi-methods.

## Limitations and strengths

This study has four types of limitations. First, the use of only verbal interviews limited the sources of data, hence the larger landscape of contextual understanding of the issues. Second, the degree to which these results could be generalized to the whole nation is unclear but warrants examination. Further studies should target women using all the levels of health facilities including private health facilities. Third, there were limitations with the translation between English and Khmer. Generally, the more limited vocabulary in Khmer compared with English created difficulties in seeking appropriate words/terms in Khmer. Both implicit and explicit cultural differences between written and oral languages were linguistic complex challenges in conducting cognitive interviewing. Fourth, it was difficult to conduct interviews with women in private. A Cambodian research assistant suggested that it was not culturally appropriate to let their family away from early postpartum women, due to women's unique perception towards privacy and confidentiality. Thus, women's answers may have been biased by social desirability when family members were present.

Nonetheless, the study has several strengths. First, the present study addressed the exploration of cultural contexts for the cross-cultural adaptation process. It contributed to achieving cultural equivalence and content validity of the tool. Second, we successfully pioneered online cognitive interviewing in the developing country during the COVID-19 pandemic.

## Conclusion

Contextual understandings contributed to more precisely adapting the PCMC scale and identifying methodological issues in developing a cross-cultural tool. This study also highlights the importance of cross-cultural translation and adaptation, before formally using the questionnaire in a survey. Translation and adaptation processes require not only technical and linguistic skills and experiences, but also considerable time necessary for ensuring cross-cultural

equivalence. A further study needs to be conducted, to assess broader landscapes and greater perspectives. Further, the use of a variety of data including anthropological approaches is key to fully capturing Cambodian women's reality.

## Supporting information

**S1 Checklist.**
(DOCX)

**S1 Table. Khmer translation revision process.**
(DOCX)

**S2 Table. Kh-PCMC scale.**
(DOCX)

## Acknowledgments

This is a part of the doctoral dissertation research. We express appreciation to Prof. Hisao Sekine, Ms. Kaoru Komi, Prof. Asako Takekuma Katsumata, and Dr. Togoobaatar Ganchimeg. We are grateful to all the people who contributed to this study at various stages in developing the Kh-PCMC scale, including translators, Cambodian experts, a field coordinator, field assistants, and study participants in Cambodia.

## Author Contributions

**Conceptualization:** Yuko Takahashi Naito, Rieko Fukuzawa, Patience A. Afulani.

**Data curation:** Yuko Takahashi Naito.

**Formal analysis:** Yuko Takahashi Naito, Rieko Fukuzawa, Patience A. Afulani.

**Funding acquisition:** Yuko Takahashi Naito, Hirotsugu Aiga.

**Investigation:** Yuko Takahashi Naito.

**Methodology:** Yuko Takahashi Naito, Rieko Fukuzawa, Patience A. Afulani.

**Project administration:** Yuko Takahashi Naito.

**Resources:** Yuko Takahashi Naito, Patience A. Afulani, Rattana Kim.

**Supervision:** Rieko Fukuzawa, Rattana Kim.

**Visualization:** Yuko Takahashi Naito.

**Writing – original draft:** Yuko Takahashi Naito.

**Writing – review & editing:** Rieko Fukuzawa, Patience A. Afulani, Hirotsugu Aiga.

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
