## [Decision Letter · Decision Letter 0]

20 Jul 2022

PONE-D-22-06231Cultural adaptation of the person-centered maternity care scale at governmental health facilities in CambodiaPLOS ONE

Dear Dr. Yuko Takahashi Naito

Thank you for submitting your manuscript to PLOS ONE. After careful consideration, we feel that it has merit but does not fully meet PLOS ONE’s publication criteria as it currently stands. Therefore, we invite you to submit a revised version of the manuscript that addresses the points raised during the review process.

We look forward to receiving your revised manuscript.

Kind regards,

Sonu Goel, MD

Academic Editor

PLOS ONE

Journal Requirements:

3.We note that you have stated that you will provide repository information for your data at acceptance. Should your manuscript be accepted for publication, we will hold it until you provide the relevant accession numbers or DOIs necessary to access your data. If you wish to make changes to your Data Availability statement, please describe these changes in your cover letter and we will update your Data Availability statement to reflect the information you provide.

Reviewers' comments:

Reviewer's Responses to Questions

**Comments to the Author**

1. Is the manuscript technically sound, and do the data support the conclusions?

Reviewer #1: Partly

Reviewer #2: Yes

2. Has the statistical analysis been performed appropriately and rigorously? 

Reviewer #1: Yes

Reviewer #2: N/A

3. Have the authors made all data underlying the findings in their manuscript fully available?

Reviewer #1: Yes

Reviewer #2: Yes

4. Is the manuscript presented in an intelligible fashion and written in standard English?

Reviewer #1: Yes

Reviewer #2: Yes

5. Review Comments to the Author

Reviewer #1: have added the comments in the manuscript, file attached

Need major revision

need to know how did the sample of 20 women was justified?

How can the scale was finalized without looking for the reliability, validity and factor analysis??

Few more comments are added in the manuscript uploaded

Reviewer #2: This is an honest study I believe. Authors have disclosed their limitations too. It is very much relevant for improving MCH service delivery at different levels of healthcare of a country. This study has thrown light up on some of the ignored areas of healthcare delivery system. This has got immense role in improving access and utilization of the public healthcare system by the citizens. This qualitative research is expected to help the national policy makers to rectify the loopholes if any.

6. PLOS authors have the option to publish the peer review history of their article (what does this mean?). If published, this will include your full peer review and any attached files.

Reviewer #1: No

Reviewer #2: **Yes: **Himadri Bhattacharjya

---

## [Author Response · Author response to Decision Letter 0]

15 Sep 2022

We are extremely grateful to the editor and reviewers for their comments on the initial version of the manuscript. We have revised the manuscript based on those useful comments. Please refer to the file labeled 'Response to Reviewers', point-by-point-based responses to the respective comments.

---

## [Decision Letter · Decision Letter 1]

12 Dec 2022

Cultural adaptation of the person-centered maternity care scale at governmental health facilities in Cambodia

PONE-D-22-06231R1

Dear Dr. Naito,

We’re pleased to inform you that your manuscript has been judged scientifically suitable for publication and will be formally accepted for publication once it meets all outstanding technical requirements.

Kind regards,

Siyan Yi, MD, MHSc, PhD

Academic Editor

PLOS ONE

Additional Editor Comments (optional):

Reviewers' comments:

Reviewer's Responses to Questions

**Comments to the Author**

1. If the authors have adequately addressed your comments raised in a previous round of review and you feel that this manuscript is now acceptable for publication, you may indicate that here to bypass the “Comments to the Author” section, enter your conflict of interest statement in the “Confidential to Editor” section, and submit your "Accept" recommendation.

Reviewer #1: All comments have been addressed

2. Is the manuscript technically sound, and do the data support the conclusions?

Reviewer #1: Yes

3. Has the statistical analysis been performed appropriately and rigorously? 

Reviewer #1: Yes

4. Have the authors made all data underlying the findings in their manuscript fully available?

Reviewer #1: Yes

5. Is the manuscript presented in an intelligible fashion and written in standard English?

Reviewer #1: Yes

6. Review Comments to the Author

Reviewer #1: Thank you for clarifying all the comments so well!

Great work, It will definitely help in improving the quality of care further.

7. PLOS authors have the option to publish the peer review history of their article (what does this mean?). If published, this will include your full peer review and any attached files.

Reviewer #1: **Yes: **Bharti Sharma

---

## [Editor Report · Acceptance letter]

23 Dec 2022

PONE-D-22-06231R1 

Cultural adaptation of the person-centered maternity care scale at governmental health facilities in Cambodia 

Dear Dr. Naito:

I'm pleased to inform you that your manuscript has been deemed suitable for publication in PLOS ONE. Congratulations! Your manuscript is now with our production department. 

Kind regards, 

on behalf of

Dr. Siyan Yi 

Academic Editor

PLOS ONE